# A Compact Cat Swarm Optimization Algorithm Based on Small Sample Probability Model

**Zeyu He, Ming Zhao \*** , **Tie Luo and Yimin Yang**

School of Computer Science, Yangtze University, Jingzhou 434025, China
\* Correspondence: hitmzhao@gmail.com

**Abstract:** In this paper, a compact cat swarm optimization algorithm based on a Small Sample Probability Model (SSPCCSO) is proposed. In the same way as with previous algorithms, there is a tracking mode and a searching mode in the processing of searching for optimal solutions, but besides these, a novel differential operator is introduced in the searching mode, and it is proved that this could greatly enhance the search ability for the potential global best solution. Another highlight of this algorithm is that the gradient descent method is adopted to increase the convergence velocity and reduce the computation cost. More importantly, a small sample probability model is designed to represent the population of samples instead of the normal probability distribution. This representation method could run with low computing power of the equipment, and the whole algorithm only uses a cat with no historical position and velocity. Therefore, it is suitable for solving optimization problems with limited hardware. In the experiment, SSPCCSO is superior to other compact evolutionary algorithms in most benchmark functions and can also perform well compared to some population-based evolutionary algorithms. It provides a new means of solving small sample optimization problems.

**Keywords:** compact cat swarm optimization; differential operator; small samples probability model; gradient descent method

## 1. Introduction

Lately, compact evolutionary algorithms (cEA) have rapidly developed. In 1999, Georges R. Harik et.al proposed a novel compact genetic algorithm (cGA) [1]. It mimics the behavior of a simple GA with standard binary coding for order-one problem. It could almost obtain the same performance as a standard GA under this simple representation. Besides these, it engendered an idea that it is possible to use a probability distribution to represent populations, in order to reduce memory usage. Inspired by cGA, Ernesto Mininno et al. proposed a real-valued Compact Genetic Algorithms (rcGA) [2]. It firstly employed a normal probabilistic distribution model to describe the statistic features of all of the samplings. Individuals could be generated directly from this normal probabilistic distribution model. The most successful highlight of rcGA lies in that it employed effective updating rules to update the parameters of the normal probabilistic distribution functions (PDF). A compact Differential Evolutionary algorithm (cDE) [3] was presented by Ernesto Mininno et al. in 2011. Though it was based on the same normal probabilistic model, it inherited the essential features of differential evolutionary (DE). The efficient performance together with modest requirements made it suitable for the environment with small computational power. After cDE, cPSO [4] was proposed in 2013. Unlike the other PSO versions, it stored neither the positions nor the velocities, and only a particle was employed in the whole algorithm; what is more, it also employed a normal probabilistic model to simulate the swarm's behavior. This modest representation enables cPSO to run in those devices with limited computational power or limited memory. In 2018, Ming Zhao [5] proposed

a novel compact cat swarm optimization algorithm based on a differential method, with better performance than some similar algorithms.

These compact evolutionary algorithms employed a probability distribution to explicitly represent the population of the solutions. Normally, a normal distribution function is introduced. Instead of a large number of populations and variables, only the expectation and variance of the representative probability model are adopted, a type of probability model and a particle are adopted and a few variables and limited run spaces are required; that is, a compact idea is used to design the algorithm. It is known to all that a good distribution is equivalent to linkage learning [6,7]. A normal distribution model suits the simulation of those samplings with a large size [8,9]. Obviously, not all of the samplings could be described by normal distribution. There are still some samplings with non-normal distribution. If we employ a normal distribution to simulate them compulsively, the performance will be barely satisfactory. Therefore, the problems with a small size could probably be described by a special non-normal distribution.

Inspired by the literature [1–5], we expected to find another non-normal probabilistic distribution model to represent the samplings, which would have a different mean and the variance under the different parameters. Updating the rules for the mean and the variance could help the probabilistic model to generate highly effective solutions. Giving overall consideration to the features of some non-normal probability models, a gamma probabilistic distribution function is employed in this study.

Meanwhile, the corresponding evolutionary algorithm will be considered. It will cooperate with the gamma probabilistic model to try to find the best solution for the optimization problems. Chu et al. [10] proposed a cat swarm optimization (CSO) algorithm in 2007. A novel combination searching strategy was employed in CSO, in comparison with the corresponding evolutionary algorithms, a higher performance was shown in the standard test functions. Then, Tsai and his group [10] developed it further and proposed some improved versions, such as parallel cat swarm optimization (PCSO) [11] and reinforced parallel cat swarm optimization (EPCSO) [12]. It was also widely used in some application domains with a pretty good performance [13–15]. It is population-based. There is still no population-less version for CSO. So, in this paper, we also select CSO to combine the gamma probabilistic distribution, which is a Small Sample Probability Model. In order to reduce computing costs and the velocity up convergence rate, a gradient descent method is introduced to the seeking mode of CSO. Thus, a novel compact cat swarm optimization scheme with a Small Sample Probability Model (SSPCCSO) is proposed. We will employ this novel algorithm to test whether it could solve some problems of a small size.

The remainder of this research is organized as follows: Section 2 presents the sampling mechanism and cat swarm optimization. In Section 3, the proposed compact cat swarm optimization with gamma distribution and gradient descent method is discussed in detail. Section 4 displays the experimental results, and Section 5 summarizes this study.

## 2. Related Work

In this section, the sampling mechanism based on real-valued coding is presented in detail, and the cat swarm optimization algorithm is also introduced.

### 2.1. The Sampling Mechanism Based on Real-Valued

As mentioned above, the main feature of compact algorithms is population-less, see [1–4]; a virtual population based on probabilistic model is introduced to represent the populations. In real-valued coding compact algorithms [2–4], solutions are generated through this probabilistic model. Mean and variance are parameters for the probabilistic density function (PDF). A little modification for the two parameters will affect the solutions in the next generation. So, this probabilistic model is named as Perturbation Vector (PV). It is encoded as Formula (1):

$$PV^t = [u^t, \delta^t] \tag{1}$$

where $\mu$ and $\sigma$ are, respectively, mean and standard deviation values of the corresponding *PDF*; top *t* is generations.

Without losing generality, all of designed variable *x* should be normalized to interval $[-1,1]$, for a *PDF*, its corresponding Cumulative Distribution Function (*CDF*) may not be equal to 1 because some of the variables will be out of $[-1,1]$. An error function [15] is introduced to solve this problem; thus, the truncated *PDF* is presented as Formula (2):

$$PDF(truncated(x_i)) = \frac{\sqrt{\frac{2}{\pi}}e^{-\frac{(x_i-u_i)^2}{2(\sigma_i)^2}}}{\sigma_i(erf(\frac{u_i+1}{\sqrt{2}\sigma_i}) - erf(\frac{u_i-1}{\sqrt{2}\sigma_i}))} \tag{2}$$

where $x_i$ is the *i*-th dimension of designed variable *x*; $u_i$ and $\delta_i$ are mean and variance associated with $x_i$. The corresponding *CDF* value can be obtained through Formula (3). When a Cumulative Distribution Function value is generated, the corresponding *x* could be calculated by computing the inverse function of Formula (3). The sampling mechanism of *PV* can be seen in Figure 1.

$$CDF(x_i) = \int\limits_{-1}^{1} PDF(truncated\ x_i)dx_i \tag{3}$$

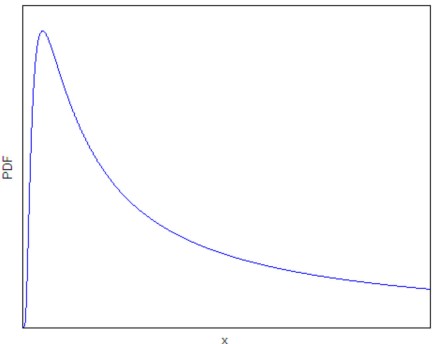 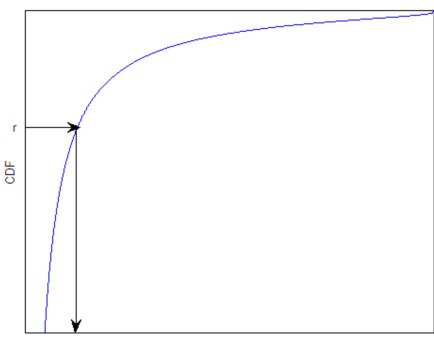

**Figure 1.** Sampling Mechanism for Real-Valued Compact.

The sampling procedure could be described as follows. Firstly, a random number in [0,1] is generated under a normal distribution, and it will be taken as a Cumulative Distribution Function value; then, computing the inverse function of the Cumulative Distribution Function, the calculated value is $x_i$. $x_i$ is a needed solution, however, the solution is not generated directly, it was obtained based on e evolutionary computation and evolutionary computation. The sampling mechanism could be interpreted as Figure 1.

In the real process of sampling, in order to reduce the calculated cost, an approximate computing for the designed *x* [*i*] is implemented, by means of the Chebyshev polynomials [16].

Another highlight for the sampling mechanism of the compact evolutionary algorithms with real-value is updating the rules for mean and variance. When two individuals are compared, the *winner* indicated the one with better fitness, and the other is *the loser*. A more effective solution would be expected to be generated from *PV* through updating mean and variance. The updating rules are shown as Formulas (4) and (5):

$$\mu^{t+1}(i) = \mu^t(i) + \frac{1}{N_p}[winner(i) - loser(i)] \tag{4}$$

$$\left[\sigma^{t+1}(i)\right]^2 = \left[\sigma^t(i)\right]^2 + \left[\mu^t(i)\right]^2 - \left[\mu^{t+1}(i)\right]^2 + \frac{1}{N_p}\left\{[winner(i)]^2 - [loser(i)]^2\right\} \tag{5}$$

$N_p$ is the size of the virtual population, and top *t* is the generations, *i* is a dimension of the designed variable *x*. The details can be seen in the literature [2–4].

### 2.2. Cat Swarm Optimization (CSO)

Inspired by the behavior of cats, Chu and Tsai [10] proposed the cat swarm optimization algorithm in 2007. A combination of two search logics is employed in this algorithm, i.e., the seeking mode and the tracing mode. All of the cats will be divided into two groups before the iterations. Just like PSO, its update rules are very similar to the traditional Particle Swarm Optimization (PSO) algorithm; the designed cat represents a solution for the project to be solved, and each designed cat has its own position and velocity. The solutions are updated by the cat's position and velocity, and estimated based on its degree of adaptability for the project; the global best will be chosen and conduct all of the cats to seek its next position and velocity. From the perspective of biological group behavior, this is obviously different from the particle swarm optimization algorithm. The details of the CSO algorithm will be introduced in the following section.

#### 2.2.1. Seeking Mode

The number of cat populations in the seeking mode is decided by a parameter GR (group rate), which, normally, is set to be 0.98 [8]. When the cats are in seeking mode, the GR is used as a minor tune-up for the cats' position, and does change their velocities. The following steps will be implemented.

Firstly, every cat will copy its own position many times, according to the size of the parameter SMP, and the position will be stored in the corresponding seeking mode pool (SMP) unit. Then, each position in the SMP will be recalculated by a mutagenic operator, a dimension of the expected variable $x_i$ could be chosen to mutate, and the range of variation would be decided by a random number, which is up to 20% of $x_i$. The mutation operation is described as Formula (6):

$$x_i = x_i + \Delta x_i \tag{6}$$

The position with the best fitness in the SMP will be chosen to update $x_i$.

#### 2.2.2. Tracing Mode

The evolutionary process for the cats in tracing mode is similar to the particle in the PSO algorithms. However, they are still somewhat different. Each cat in tracing mode will only trace the cat with the global best fitness to update its own velocity and position. The particles in PSO [16] trace both the global best individual and the local best individual. The updating rule for the tracing mode can be expressed as the Formulas (7) and (8):

$$\text{v}_k(t+1) = \omega \cdot \text{v}_k(t) + Const \cdot random \cdot [x_{gb}(t) - x_k(t)] \tag{7}$$

$$x_k(t+1) = x_k(t) + v_k(t+1) \tag{8}$$

where $x_{gb}$ is the position of the cat with the best fitness; $x_k$ is the position of $cat_k$; $t$ is the generation for iterations. *Const* is a constant and *random* is a random number in [0,1].

The cat in seeking mode will be compared with the cat with the best fitness in tracing mode; the winner would be chosen to update the variable. The final $x_{gb}$ is the required solution.

### 3. The Proposed Compact Cat Swarm Optimization Scheme Based on Small Sample Probability Model (SSPCCSO)

In this section, a novel compact swarm optimization scheme based on the Small Sample Probability Model will be proposed. First, a sampling mechanism with a new gamma distribution model will be introduced in Section 3.1, then a new differential operation will be implemented in seeking mode. Another highlight, a gradient descent method, will also be presented in Section 3.2. Section 3 states the tracing mode, and Section 4 introduces the SSPCCSO.

### 3.1. Virtual Population and Sampling Mechanism with Real-Valued

The main feature of the compact evolutionary optimization algorithms is population-less. A probabilistic model is employed to represent the distribution of the solution sets, instead of processing an actual population. A gamma distribution model is employed to act as the Perturbation Vector (*PV*). The *PV* is also a $n \times 2$ matrix and it is expressed as Formula (1). As is mentioned above, the *PV* is introduced to generate a new individual. The sampling mechanism is the same as the PV with a normal distribution (see Figure 1).

Normally, a gamma probability density function (*PDF*) and its *CDF* [17] are presented as Formula (9) or (10):

$$f(x; k, \theta) = \frac{1}{\Gamma(k)\theta^k} x^{k-1} e^{-x/\theta} \tag{9}$$

$$F(x; k, \theta) = \int_0^x f(u; k, \theta) du = \frac{1}{\Gamma(k)} \gamma(k, \frac{x}{\theta}) \tag{10}$$

where $k$ and $\theta$ are two parameters for Gamma *PDF*; the *PDF* curve is shown as Figure 2 [13].

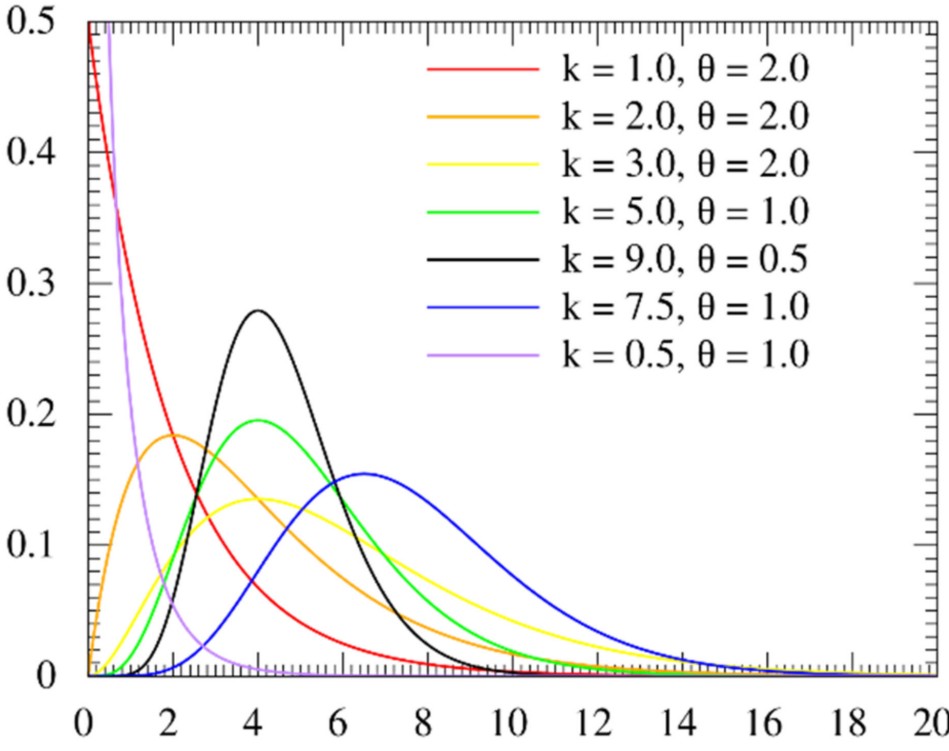

**Figure 2.** Gamma Probability Density Function.

The lower incomplete Gamma function is defined by $\gamma(s, x) = \int_0^x t^{s-1} e^{-t} dt$ (note that the upper incomplete Gamma function is $\Gamma(s, x) = \int_x^\infty t^{s-1} e^{-t} dt$, and the ordinary Gamma function is defined as $\Gamma(s) = \Gamma(s, 0) = \gamma(s, \infty)$). Thus, the mean and variance for gamma *PDF* is calculated by $E[X] = k\theta$ and $Var[X] = k\theta^2$.

Most of the variables are in the interval [0,20]. We define [0,20] as an all solution domain. However, there may still be some potential solutions out of [0,20], so an error function must be employed to map those potential solutions out of [0,20] into [0,20].

Without losing generality, all of the variables should be normalized to $[-1,1]$. As discussed above, we define the variables $x$ in $[0,20]$, then new variable will be mapped as $y = 10(x+1)$. Thus, we consider the truncated Gamma distribution to $[0,20]$, and then:

$$
\begin{aligned}
f(10(x+1);k,\theta) &= \frac{10^{k-1}}{\Gamma(k)\theta^k}(x+1)^{k-1}e^{-10(x+1)/\theta} \\
&= \frac{1}{10\Gamma(k)\left(\frac{\theta}{10}\right)^k}(x+1)^{k-1}e^{-(x+1)/\left(\frac{\theta}{10}\right)} \\
&= \frac{1}{10\Gamma(k)t^k}(x+1)^{k-1}e^{-(x+1)/t} \\
&= \frac{1}{10}f(x+1;k,t)
\end{aligned}
\tag{11}
$$

$$
F(10(x+1);k,\theta) = \frac{1}{\Gamma(k)}\gamma\left(k,\frac{10(x+1)}{\theta}\right) = \frac{1}{\Gamma(k)}\gamma\left(k,\frac{x+1}{t}\right)
\tag{12}
$$

Thus, the new distribution truncated on $[-1,1]$ is represented by Formula (13):

$$
\begin{aligned}
PDF(truncated) &= \frac{1}{10}\frac{f(x+1;k,\theta)}{\frac{1}{\Gamma(k)}\gamma\left(k,\frac{2}{\theta}\right)} \\
&= \frac{1}{10\gamma\left(k,\frac{2}{\theta}\right)\theta^k}(x+1)^{k-1}e^{-(x+1)/t}
\end{aligned}
\tag{13}
$$

Then, it could ensure that any of the solutions out of $[-1,1]$ could map to $[-1,1]$, see Figure 3. Its *CDF* is presented by Formula (14):

$$
\begin{aligned}
CDF(truncated) &= \frac{\frac{1}{\Gamma(k)}\gamma\left(k,\frac{x+1}{\theta}\right)}{\frac{1}{\Gamma(k)}\gamma\left(k,\frac{2}{\theta}\right)} \\
&= \frac{\gamma\left(k,\frac{x+1}{\theta}\right)}{\gamma\left(k,\frac{2}{\theta}\right)}
\end{aligned}
\tag{14}
$$

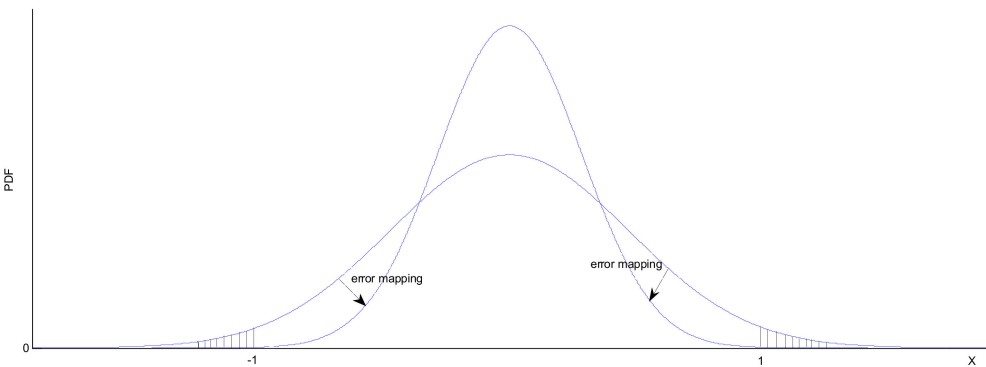

**Figure 3.** Error mapping for some solutions out of the decision domain.

According to Formulas (13) and (14), the mean and variance could be represented by $\mu = \frac{k\theta}{10}$ and $\sigma = \frac{k\theta^2}{100}$.

The sampling mechanism of a designed variable could be described as follows:

First, the scheme will generate a random number $p$ between 0 and 1, according to the uniform distribution model, and the parameters $\mu_i$ and $\sigma_i$ for the Perturbation Vector will be Initialized ($\mu_i = 0$ and $\sigma_i = \lambda$). This $p$ is the Cumulative Distribution Function (CDF) value for the expected variable $x$, it is *a* Cumulative Distribution Function value for the corresponding *PDF*, then the inverse function of *CDF* in rand (0, 1) is introduced, according to Formula (14). Finally, a new $x[i]$ will be obtained.

Apparently, according to the definition of normal distribution, its solutions' domain should be $[-\infty, +\infty]$, but real projects are limited to a specific domain, The optimal solution obtained by updating the rules may be not located in the definition domain, an error will be generated for mapping an infinite space to a finite space; the mapping function is introduced to solve the error for mapping. Because of the particularity of the Cumulative Distribution Function for Gamma distribution, all of the solutions almost locate in $[0,20]$.

The mapping problem turns out to be a finite domain into another finite domain, the error disappears automatically, thus, an error function is not required.

The Perturbation Vector Updating Rule

The *PV* of the virtual population is designed to create new solutions; the parameters of *PV* could be updated to create more significant individuals. The updating rules for *PV* are also the same as in the literature [2–4]. They are expressed as Formulas (4) and (5). There are two very important vectors, *winner* and *loser*, in Formulas (4) and (5), in which *winner* indicates the individual with best fitness when two solutions are compared. From Formulas (4) and (5), $\mu$ and $\sigma$ are updated by an vector $\frac{1}{N_p}(winner - loser)$, this vector could adjust the values of $\mu$ and $\sigma$. Apparently, the new vector $\frac{1}{N_p}(winner - loser)$ conducts the forward orientation of $\mu$; thus, the $\mu$ would approach the *winner*, in order to obtain the next solution more effectively; $\sigma$ is designed to conduct the step size, when the current solution is far from the best solution, a large size $\sigma$ could be used. When the current solution is close to the best solution, a small $\sigma$ would be used. Thus, a new solution in the next iteration would be generated more effectively by this updating rule. This can be seen in Figure 4.

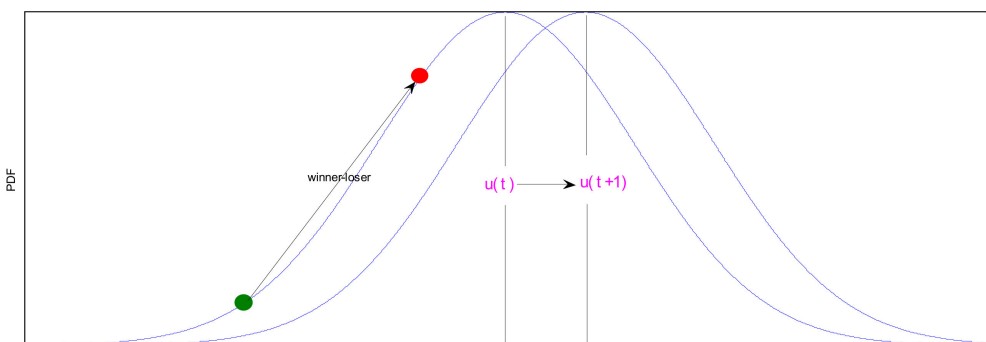

**Figure 4.** The interpretation for updating rule with *winner* and *loser*.

### *3.2. Seeking Mode*

Compared with CSO, the SSPCCSO has two highlights: one is the differential operator, and the other is the gradient descent method. The details about these will be presented in this section.

### 3.2.1. Differential Operator

The cat in seeking mode will update its position by another new way. For the seeking mode of CSO, the position of a cat will be updated through Formula (6), while in SSPCCSO, a differential operator is introduced to enhance the search ability of the cat for the local best solution, and we call this differential operator the Exploration Vector (*EV*), it is presented as Formula (15):

$$x = x + F * (winner - loser) \tag{15}$$

where $F \in [1/N_p, 0.2]$ is a scale factor which controls the length of the exploration vector $(winner - loser)$.

It had to re-mention the vector $(winner - loser)$, the updating rule for seeking in the CSO is described as Formula (6); according to Formula (6), $x$ will be changed from the view of $x$ itself, that is to say, the updating vector $(x + \Delta x)$ is only a simple amplification or reduction in the scalar on its direction. It can find the better solution based on two directions, it can be seen on the left of Figure 5, according to the proposed Formula (15), where it could search for the best solution in all directions. It can also be seen on the right of Figure 5.

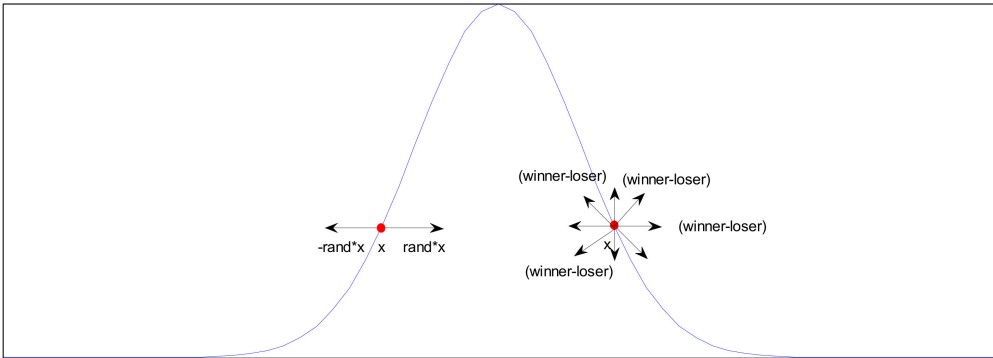

**Figure 5.** The search direction for different updating rules.

(*Winner-loser*) is an exploration vector, it could face in all of the potential directions, and it has challenges in seeking for a better solution, see on the right of Figure 5.

With reference to the literature [10–12], the proportion for cats in the seeking mode and cats in the tracing mode is 98:2, and the size of memory pool is five times of the current cat. However, in the proposed SSPCCSO, a cat is only in seeking mode or in tracing mode. In order to mimic the search logic of the CSO algorithm, the updating rule would be implemented 245 times in each iteration; thus, the computing cost may be too high to be accepted. A gradient descent method [18,19] is introduced to reduce the computing cost and obtain the real local solution for the designed *x*. The details will be discussed in the following section.

### 3.2.2. Gradient Descent Method

As mentioned in the previous section, the updating rule in seeking mode will be implemented many times, and the process will be run with a higher computing cost. It may not be accepted for some of the engineering problems. With full consideration of these factors, a gradient descent method (GD) [19,20] is introduced. Firstly, a convergence rate (CR) is shown as Formula (16):

$$CR = \frac{|fitness(t+2) - fitness(t+3)|}{|fitness(t) - fitness(t+1)|} \tag{16}$$

where $fitness(t)$ is the fitness for the cat in $t$ generation. When $CR < 1$, it means that the acquired solution begins to converge [20].

When the selection for the local best solution is being implemented, and the termination condition is not met, and meanwhile the local best cat is not updated for many generations, the rest loop would be unnecessary. Even though the local best solution is frequently to obtain a differential operator, sometimes this local best solution is not actually the local best solution. Based on these factors, much unnecessary computing cost could be reduced, and a more effective solution could be found with less time. Thus, a gradient is introduced and is defined as Formula (17).

$$\nabla f = [\frac{\partial f}{\partial x_1}, \frac{\partial f}{\partial x_2}, \cdots, \frac{\partial f}{\partial x_n}] \tag{17}$$

where $n$ is the dimension of designed variable $x$. It is known to all that the real local best solution of variable $x$ will be quickly obtained, according to the gradient vector of the function, and it can be calculated by Formula (18):

$$x(t+1) = x(t) + \nabla f \cdot d \tag{18}$$

where $d$ is the step length, and it could firstly be set as the vector (*winner-loser*), then it can be optimized by processing $max(f(x_i + \nabla f \cdot d)$. The flow chart of the steepest gradient descent method is shown as Figure 6.

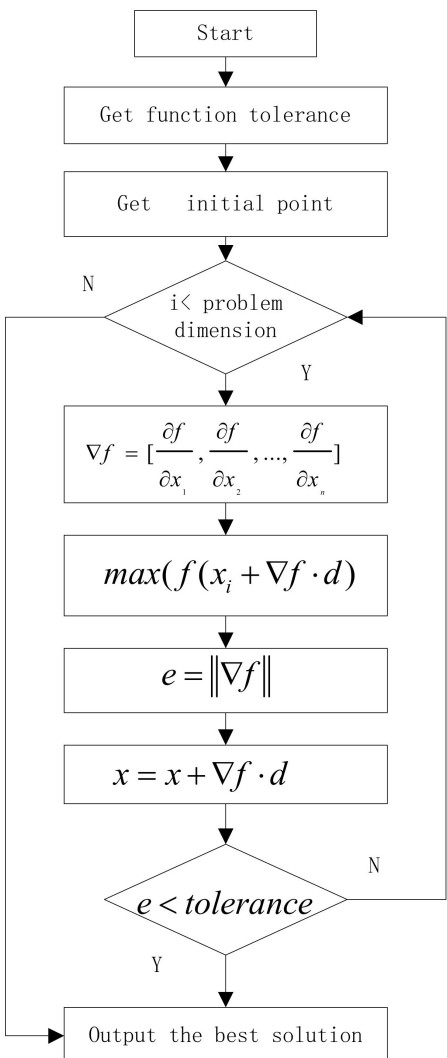

**Figure 6.** The flow chart for gradient descent method in seeking mode.

When the evolutionary procedure of the designed $x$ goes into the gradient descent phase, the computing cost will be highly reduced, and the real local best solution will be quickly obtained.

### 3.3. Tracing Mode

The cat's behavior in tracing mode is simulated to the particle in the PSO algorithm, but each cat only traces the cat with the global best position to update its own velocity and position. The updating rule of tracing mode could be presented as Formulas (7) and (8). All of the parameters are the same to the original CSO [9].

The combination of seeking mode and tracing mode could ensure that the cat swarm optimization algorithm converges quickly and prevents the solution from the local optimum.

A cat with the best fitness will be chosen to compare which is the final best solution. When the iterations meet the termination condition, the final $xgb$ is the solution for the problem.

### 3.4. The Procedure and Pseudo Code for SSPCCSO

For the proposed SSPCCSO, normally, all of the variables would be mapped into the intervals $[-1,1]$, $\mu$ and $\sigma$ are initialized as $\mu_i = 0$ and $\sigma_i = \lambda$, a random number in the range $[-1,1]$ is chosen as the global best $xgb$. Then, the cat would be randomly grouped into a mode.

In the iteration phase, when the cat is in tracing mode, a local best solution $x_{lb}$ is generated from $PV$, the cat's position and velocity is updated by Formulas (7) and (8), and the comparison between $x_{lb}$ and $cat.x^{t+1}$ is used to determine which is *winner* and which is *loser*. Then, *winner* and *loser* are applied to updating the $PV$. Another comparison between the $cat.x^{t+1}$ and $xgb$ is used to decide who is $xgb$. When the cat is in seeking mode, firstly, a new candidate solution $x_{lb}$ from $PV$ would compare with $cat.x$; it is also employed to determine the *winner* and *loser*, as *winner* and *loser* in this case are applied to update $cat.x^{t+1}$ by Formula (12) and $PV$. These steps would be reduplicated for many times in a loop. When this loop is stagnation, a gradient descent method is involved to reduce the computing cost and find the real local best. No matter which mode the cat is in, a $xgb$ would be chosen in each run. For the sake of clarity, the pseudo code for SSPCCSO is shown in Figure 7.

```
temp=0; mean=0 and delta =10;

random generate xgbest, singlecat.x and singlecat. v

temp=randnumber; if (temp>0.02) singlecat. mode=smode else singlecat. mode=tmode;

while (t<Maxiteration)

if (singlecat. mode=smode)

    {generate xlbest from PVector

    singlecat.x(t+1) =singlecat.x(t)+c1*rand*(singlecat.x(t)-xlbest)

    [winner, loser] =compete(singlecat.x(t+1), xlbest);

    updating u and delta according to Formula (4) and (5)

    Singlecat.x=winner;

    [winner, loser] =competition (singlecat.x, xgbest);

        If (CR<1)

        {Gradient descent search (xgbest =max ( x + ∇f · d   )

        else xgb=winner;}

    else {generate xlbest from PVector

    singlecat. v(t+1) =w* singlecat. v(t)+c2*r2*(xgbest -singlecat.x(t)); Singlecat.x=singlecat.x+
singlecat.v

        [winner, loser] =compete(singlecat.x(t+1), xlbest);

    updating mean and delta according to Formula (4) and (5)

    [winner, loser] =compete(singlecat.x(t+1), xgbest);

    xgbest=winner;}

        temp=temp+1;

end whileloop
```

**Figure 7.** The pseudo-code for SSPCCSO.

## 4. Experimental Results and Analysis

With reference to the literature [4,20], the SSPCCSO would be tested on 47 benchmark functions, which include those where their coordinates had been transformed and their shifted [21]. The composition test functions [22] are also introduced; all of the benchmark functions are listed in the Appendix A.

SSPCCSO is a compact optimization algorithm based on small size samplings. Firstly, we select rcGA, cDE and cPSO as the compared algorithms. As SSPCCSO is also a member of evolutionary algorithms, traditional DE [23], PSO [24] and CSO [9] should be considered. From the perspective of saving memory, ISPO [25] is an indispensable object; it should be one of the objects of comparison with SSPCCSO.

All of the experiments were carried out on a personal computer with MATLAB language, which is equipped with Pentium (R) dual core E6600 CPU, 3.06 GHz and 2.96 gb RAM. The operation system is set at windows XP platform. Each comparison algorithm will select a set of parameters through which the best results can be obtained. With reference to the literature [4,26], all of the parameters for each compared algorithm are listed in Table 1. For all of the real population-based algorithms, the population size is 60, for all of the virtual population-based algorithms, the population size is 300. To achieve a truly fair comparison for all of the compared algorithms, all of the algorithms involved in the comparison were evaluated by taking the average value after running for over 30 times. In all of the tables, each value represents the corresponding mean value and the standard deviation value calculated for each comparison algorithm within 30 times, and "+", "−" and "=" have the same implication as in the literature [2].

**Table 1.** Selected parameters list for all compared algorithms in this projection.

| Algorithm | Parameters | Literature | Algorithm | Parameters | Literature |
|---|---|---|---|---|---|
| rcGA | $N_p = 300$ | [1] | DE | $N_p = 60, F = 0.5, Cr = 0.9$ | [23] |
| cDE | $N_p = 300, F = 0.5$ $Cr = 0.3$ | [2] | PSO | $\phi_1 = -0.2, \phi_2 = -0.07, \phi_3 = 3.74$ $\gamma_1 = \gamma_2 = 1, N_p = 60$ | [24] |
| cPSO | $\phi_1 = -0.2, \phi_2 = -0.07, \phi_3 = 3.74$ $\gamma_1 = \gamma_2 = 1, N_p = 300$ | [3] | ISPO | $A = 1, P = 10, B = 2, S_f = 4$ $H = 30, \varepsilon\varepsilon = 1.0 \times 10^{-5}$ | [25] |
| CSO | $N_p = 60, c1 = c2 = 2$ $W = 0.9$ | [10] | SSPCCSO | $w = -0.4, c1 = 2, c2 = -0.07$ $N_p = 300$ | |

The remains of this section are described as below: first, a comparison for memory usage is listed in Table 2. Then, the comparisons for the memory-saving algorithms are presented. Next to this, the comparisons between the population-based algorithms and SSPCCSO will be shown; the analyses of the results for SSPCCSO are summarized in the final section.

**Table 2.** Running memory space for all compared algorithms.

| Algorithm | Components | Memory Slots |
|---|---|---|
| ISPO | Single individual, 1 global best | 2 |
| rcGA | One individual, persistent elitism, 1 sampling | 4 |
| cDE | 3 sampling, 1 global best | 4 |
| cPSO | 1 sampling, 5 persistent variables | 5 |
| SSPCCSO | The same to cPSO | 5 |
| PSO | Population-based, history and current individuals | $2N_P$ |
| DE | Population-based, current individuals only | $N_P$ |
| CSO | Population-based, history and current individuals | $2N_P$ |

### 4.1. Comparison for Memory Usage

The proposed SSPCCSO scheme adopts a gamma probability distribution model to represent the population, and only a cat is used. The cat has the same data structure as the particle of cPSO, so it will also have the same memory space as the cPSO, in which only five persistent variables are required for storing the whole algorithm. The memory usage status for all of the compared algorithms can be seen in Table 2.

From the data in Table 2, it can be seen that SSPCCSO, cPSO, cDE, rcGA and ISPO have modest memory requirements. They belong to compact optimization algorithms. In relation to memory usage, the SSPCCSO is better than the CSO which is population-based.

### 4.2. Comparisons for Compact Optimization Algorithms

The experimental data in Table 3 display the results for the SSPCCSO and other compared compact bio-inspired algorithms. In all of the experimental results of the 47 test functions, the SSPCCSO exhibited quite good performance. Compared with the cPSO, SSPCCSO is outperformed on 30 test functions, on the contrary, the cPSO exceeds SSPCCSO only on 17 benchmark functions. Among all of the memory-saving algorithms, SSPCCSO is out-performed by the other compared algorithms over 24 functions. From the view of the mathematics method, it lies in two factors, first of all, a differential operator in seeking mode is introduced to substitute for the original mutation operator. The difference between a solution $cat.x[i]$ and another variable may generate a moving direction, it may be a 360 degree angle transformation for the existing solution; the magnitude for the $cat.x[i]$ variation is decided by the size of the vector $(winner - loser)$. The potentially more efficient solution around $cat.x[i]$ will be found according to this searching method, which was similar to the cDE algorithm. Secondly, SSPCCSO also kept the search logic of the PSO, that is to say, the proposed SSPCCSO has the search ability of both the cPSO and cDE. It combines these two searching abilities; thus, it is not surprising that its search performance exceeds these two algorithms.

**Table 3.** Comparison for memory-saving algorithms.

| Function | rCGA | cDE | ISPO | cPSO | W | SSPCCSO |
|---|---|---|---|---|---|---|
| fu1 | $1.427 \times 10^4 \pm$ $9.27 \times 10^3$ | $8.73 \times 10^{-28} \pm$ $1.86 \times 10^{-28}$ | $8.437 \times 10^{-31} \pm$ $3.31 \times 10^{-31}$ | $6.471 \times 10^1 \pm$ $2.28 \times 10^1$ | + | $\mathbf{6.170 \times 10^{-3} \pm}$ $\mathbf{1.05 \times 10^{-3}}$ |
| fu2 | $2.851 \times 10^4 \pm$ $6.58 \times 10^3$ | $3.778 \times 10^3 \pm$ $1.85 \times 10^3$ | $1.184 \times 10^1 \pm$ $5.92 \times 10^0$ | $2.560 \times 10^3 \pm$ $2.37 \times 10^3$ | + | $\mathbf{3.625 \times 10^2 \pm}$ $\mathbf{7.02 \times 10^2}$ |
| fu3 | $1.282 \times 10^9 \pm$ $1.58 \times 10^9$ | $1.291 \times 10^2 \pm$ $1.84 \times 10^2$ | $2.026 \times 10^2 \pm$ $3.28 \times 10^2$ | $1.320 \times 10^5 \pm$ $7.46 \times 10^4$ | + | $\mathbf{5.776 \times 10^{-1} \pm}$ $\mathbf{7.01 \times 10^0}$ |
| fu4 | $1.874 \times 10^1 \pm$ $3.59 \times 10^{-1}$ | $8.694 \times 10^{-2} \pm$ $2.97 \times 10^{-1}$ | $1.942 \times 10^1 \pm$ $1.57 \times 10^{-1}$ | $3.728 \times 10^0 \pm$ $3.71 \times 10^{-1}$ | + | $\mathbf{5.574 \times 10^{-1} \pm}$ $\mathbf{3.07 \times 10^{-2}}$ |
| fu5 | $6.434 \times 10^{-3} \pm$ $1.31 \times 10^{-2}$ | $4.289 \times 10^{-3} \pm$ $1.38 \times 10^{-2}$ | $1.124 \times 10^1 \pm$ $1.77 \times 10^1$ | $\mathbf{9.63 \times 10^{-8} \pm}$ $\mathbf{3.07 \times 10^{-8}}$ | − | $1.613 \times 10^{-2} \pm$ $2.37 \times 10^{-3}$ |
| fu6 | $1.963 \times 10^2 \pm$ $2.85 \times 10^1$ | $7.944 \times 10^1 \pm$ $1.48 \times 10^1$ | $2.548 \times 10^2 \pm$ $4.23 \times 10^1$ | $2.94 \times 10^1 \pm$ $7.94 \times 10^0$ | + | $\mathbf{2.399 \times 10^{-2} \pm}$ $\mathbf{4.09 \times 10^{-3}}$ |
| fu7 | $2.312 \times 10^3 \pm$ $2.47 \times 10^3$ | $4.983 \times 10^3 \pm$ $3.78 \times 10^3$ | $2.254 \times 10^3 \pm$ $8.62 \times 10^2$ | $4.614 \times 10^2 \pm$ $2.40 \times 10^2$ | + | $\mathbf{2.991 \times 10^2 \pm}$ $\mathbf{1.75 \times 10^{-1}}$ |
| fu8 | $3.194 \times 10^3 \pm$ $8.01 \times 10^2$ | $1.673 \times 10^3 \pm$ $4.48 \times 10^2$ | $5.768 \times 10^3 \pm$ $5.38 \times 10^2$ | $3.160 \times 10^3 \pm$ $9.75 \times 10^2$ | + | $\mathbf{1.248 \times 10^1 \pm}$ $\mathbf{1.90 \times 10^0}$ |
| fu9 | $\mathbf{1.008 \times 10^4 \pm}$ $\mathbf{2.35 \times 10^3}$ | $8.548 \times 10^3 \pm$ $2.14 \times 10^3$ | $2.755 \times 10^4 \pm$ $6.08 \times 10^3$ | $1.344 \times 10^4 \pm$ $1.74 \times 10^3$ | − | $1.111 \times 10^5 \pm$ $5.29 \times 10^4$ |
| fu10 | $3.697 \times 10^5 \pm$ $1.78 \times 10^5$ | $4.265 \times 10^4 \pm$ $2.35 \times 10^4$ | $\mathbf{4.326 \times 10^3 \pm}$ $\mathbf{4.54 \times 10^3}$ | $1.040 \times 10^6 \pm$ $1.16 \times 10^5$ | + | $9.390 \times 10^5 \pm$ $1.13 \times 10^4$ |
| fu11 | $1.851 \times 10^1 \pm$ $4.37 \times 10^{-1}$ | $1.708 \times 10^0 \pm$ $1.11 \times 10^0$ | $1.948 \times 10^1 \pm$ $1.89 \times 10^{-1}$ | $3.699 \times 10^0 \pm$ $3.53 \times 10^{-1}$ | + | $\mathbf{8.328 \times 10^{-2} \pm}$ $\mathbf{8.23 \times 10^{-2}}$ |

**Table 3.** *Cont.*

| Function | rCGA | cDE | ISPO | cPSO | W | SSPCCSO |
|---|---|---|---|---|---|---|
| fu12 | $5.769 \times 10^{-2} \pm$ $1.05 \times 10^{-1}$ | $2.395 \times 10^{-1} \pm$ $2.03 \times 10^{-1}$ | $\mathbf{0.001 \times 10^{-1} \pm}$ $\mathbf{0.01 \times 10^{0}}$ | $9.567 \times 10^{-8} \pm$ $2.69 \times 10^{-8}$ | + | $1.018 \times 10^{-8} \pm$ $2.61 \times 10^{-9}$ |
| fu13 | $2.154 \times 10^{2} \pm$ $3.96 \times 10^{1}$ | $\mathbf{1.314 \times 10^{2} \pm}$ $\mathbf{1.87 \times 10^{1}}$ | $2.566 \times 10^{2} \pm$ $4.15 \times 10^{1}$ | $3.924 \times 10^{1} \pm$ $2.31 \times 10^{1}$ | − | $2.70 \times 10^{2} \pm$ $1.81 \times 10^{-5}$ |
| fu14 | $3.246 \times 10^{1} \pm$ $4.53 \times 10^{0}$ | $\mathbf{2.988 \times 10^{1} \pm}$ $\mathbf{3.47 \times 10^{0}}$ | $4.777 \times 10^{1} \pm$ $4.34 \times 10^{0}$ | $3.943 \times 10^{1} \pm$ $1.15 \times 10^{0}$ | − | $7.142 \times 10^{2} \pm$ $2.37 \times 10^{-1}$ |
| fu15 | $5.251 \times 10^{0} \pm$ $5.19 \times 10^{0}$ | $\mathbf{2.315 \times 10^{-16} \pm}$ $\mathbf{5.65 \times 10^{-16}}$ | $1.184 \times 10^{-6} \pm$ $2.89 \times 10^{-17}$ | $1.778 \times 10^{0} \pm$ $4.27 \times 10^{-1}$ | + | $9.427 \times 10^{-3} \pm$ $6.15 \times 10^{-3}$ |
| fu16 | $\mathbf{-1.001 \times 10^{2} \pm}$ $\mathbf{4.43 \times 10^{-9}}$ | $\mathbf{-1.001 \times 10^{2} \pm}$ $\mathbf{1.63 \times 10^{-9}}$ | $\mathbf{-1.001 \times 10^{2} \pm}$ $\mathbf{8.38 \times 10^{-15}}$ | $\mathbf{-1.001 \times 10^{2} \pm}$ $\mathbf{8.45 \times 10^{-5}}$ | = | $\mathbf{-1.001 \times 10^{2} \pm}$ $\mathbf{0.00 \times 10^{0}}$ |
| fu17 | $1.452 \times 10^{0} \pm$ $1.88 \times 10^{0}$ | $\mathbf{2.817 \times 10^{-23} \pm}$ $\mathbf{3.16 \times 10^{-23}}$ | $9.994 \times 10^{-1} \pm$ $1.56 \times 10^{0}$ | $1.702 \times 10^{0} \pm$ $7.08 \times 10^{-1}$ | + | $9.518 \times 10^{-5} \pm$ $1.57 \times 10^{-6}$ |
| fu18 | $-5.485 \times 10^{-1} \pm$ $1.11 \times 10^{0}$ | $\mathbf{-1.150 \times 10^{0} \pm}$ $\mathbf{4.98 \times 10^{-16}}$ | $-2.258 \times 10^{-1} \pm$ $1.28 \times 10^{0}$ | $-1.030 \times 10^{0} \pm$ $7.56 \times 10^{-1}$ | − | $-4.104 \times 10^{-1} \pm$ $8.97 \times 10^{-4}$ |
| fu19 | $4.338 \times 10^{2} \pm$ $4.75 \times 10^{1}$ | $2.603 \times 10^{2} \pm$ $3.04 \times 10^{1}$ | $4.044 \times 10^{2} \pm$ $4.15 \times 10^{1}$ | $\mathbf{4.403 \times 10^{1} \pm}$ $\mathbf{3.44 \times 10^{1}}$ | − | $4.500 \times 10^{2} \pm$ $2.60 \times 10^{-3}$ |
| fu20 | $-1.517 \times 10^{1} \pm$ $2.76 \times 10^{0}$ | $-3.347 \times 10^{1} \pm$ $1.87 \times 10^{0}$ | $\mathbf{-3.348 \times 10^{1} \pm}$ $\mathbf{1.64 \times 10^{0}}$ | $-2.063 \times 10^{1} \pm$ $2.33 \times 10^{0}$ | − | $-1.988 \times 10^{1} \pm$ $2.33 \times 10^{-1}$ |
| fu21 | $8.372 \times 10^{3} \pm$ $1.62 \times 10^{3}$ | $5.343 \times 10^{3} \pm$ $8.47 \times 10^{2}$ | $9.679 \times 10^{3} \pm$ $1.09 \times 10^{3}$ | $4.784 \times 10^{3} \pm$ $1.09 \times 10^{3}$ | − | $\mathbf{1.42 \times 10^{0} \pm}$ $\mathbf{2.26 \times 10^{0}}$ |
| fu22 | $2.014 \times 10^{1} \pm$ $1.48 \times 10^{-1}$ | $1.787 \times 10^{1} \pm$ $2.89 \times 10^{-1}$ | $1.951 \times 10^{1} \pm$ $7.51 \times 10^{-2}$ | $3.899 \times 10^{-1} \pm$ $5.19 \times 10^{-1}$ | + | $\mathbf{7.139 \times 10^{-2} \pm}$ $\mathbf{4.17 \times 10^{-2}}$ |
| fu23 | $1.645 \times 102 \pm$ $2.36 \times 10^{1}$ | $4.042 \times 10^{1} \pm$ $1.41 \times 10^{1}$ | $\mathbf{1.247 \times 10^{-13} \pm}$ $\mathbf{1.01 \times 10^{-14}}$ | $4.657 \times 10^{-2} \pm$ $2.39 \times 10^{-2}$ | = | $\mathbf{7.319 \times 10^{-2} \pm}$ $\mathbf{5.17 \times 10^{-3}}$ |
| fu24 | $8.488 \times 10^{4} \pm$ $8.14 \times 10^{3}$ | $2.941 \times 10^{3} \pm$ $1.59 \times 10^{3}$ | $\mathbf{1.252 \times 10^{-30} \pm}$ $\mathbf{3.09 \times 10^{-31}}$ | $6.918 \times 10^{-2} \pm$ $2.54 \times 10^{-2}$ | − | $8.961 \times 10^{-3} \pm$ $1.46 \times 10^{-2}$ |
| fu25 | $-6.349 \times 10^{-3} \pm$ $3.24 \times 10^{-4}$ | $-9.161 \times 10^{-3} \pm$ $6.27 \times 10^{-4}$ | $-4.551 \times 10^{-3} \pm$ $3.79 \times 10^{-4}$ | $\mathbf{-7.85 \times 10^{-1} \pm}$ $\mathbf{1.60 \times 10^{-14}}$ | − | $0.000 \times 10^{0} \pm$ $0.00 \times 10^{0}$ |
| fu26 | $-2.178 \times 10^{1} \pm$ $3.09 \times 10^{0}$ | $-4.938 \times 10^{1} \pm$ $3.54 \times 10^{0}$ | $\mathbf{-6.556 \times 10^{1} \pm}$ $\mathbf{3.18 \times 10^{0}}$ | $-2.920 \times 10^{1} \pm$ $2.53 \times 10^{0}$ | = | $-3.970 \times 10^{1} \pm$ $1.52 \times 10^{-1}$ |
| fu27 | $2.524 \times 10^{5} \pm$ $2.58 \times 10^{4}$ | $1.051 \times 10^{4} \pm$ $6.31 \times 10^{3}$ | $\mathbf{3.498 \times 10^{-30} \pm}$ $\mathbf{8.75 \times 10^{-31}}$ | $2.217 \times 10^{-2} \pm$ $4.04 \times 10^{-3}$ | − | $1.033 \times 10^{-2} \pm$ $1.39 \times 10^{-2}$ |
| fu28 | $1.166 \times 10^{3} \pm$ $7.35 \times 10^{1}$ | $4.218 \times 10^{2} \pm$ $3.72 \times 10^{1}$ | $7.942 \times 10^{2} \pm$ $7.69 \times 10^{1}$ | $8.776 \times 10^{-3} \pm$ $2.88 \times 10^{-3}$ | = | $\mathbf{6.631 \times 10^{-3} \pm}$ $\mathbf{1.24 \times 10^{-4}}$ |
| fu29 | $6.906 \times 10^{10} \pm$ $1.38 \times 10^{10}$ | $5.643 \times 10^{8} \pm$ $4.98 \times 10^{8}$ | $3.503 \times 10^{2} \pm$ $3.91 \times 10^{2}$ | $1.220 \times 10^{2} \pm$ $2.81 \times 10^{1}$ | + | $\mathbf{1.286 \times 10^{0} \pm}$ $\mathbf{2.50 \times 10^{0}}$ |
| fu30 | $1.297 \times 10^{11} \pm$ $2.63 \times 10^{10}$ | $7.066 \times 10^{10} \pm$ $1.19 \times 10^{10}$ | $9.702 \times 10^{9} \pm$ $3.26 \times 10^{9}$ | $\mathbf{4.928 \times 10^{6} \pm}$ $\mathbf{6.56 \times 10^{5}}$ | − | $1.109 \times 10^{8} \pm$ $3.44 \times 10^{8}$ |
| fu31 | $2.148 \times 10^{4} \pm$ $2.51 \times 10^{3}$ | $1.842 \times 10^{4} \pm$ $1.29 \times 10^{3}$ | $1.971 \times 10^{4} \pm$ $1.28 \times 10^{3}$ | $\mathbf{1.045 \times 10^{4} \pm}$ $\mathbf{2.94 \times 10^{3}}$ | − | $4.989 \times 10^{4} \pm$ $8.38 \times 10^{4}$ |
| fu32 | $1.591 \times 10^{3} \pm$ $1.27 \times 10^{3}$ | $1.062 \times 10^{-5} \pm$ $9.78 \times 10^{-6}$ | $\mathbf{2.684 \times 10^{-30} \pm}$ $\mathbf{4.75 \times 10^{-31}}$ | $1.531 \times 10^{-2} \pm$ $3.80 \times 10^{-3}$ | = | $1.774 \times 10^{-2} \pm$ $2.89 \times 10^{-2}$ |
| fu33 | $1.258 \times 10^{2} \pm$ $6.44 \times 10^{0}$ | $8.948 \times 10^{1} \pm$ $6.18 \times 10^{0}$ | $1.773 \times 10^{2} \pm$ $5.91 \times 10^{0}$ | $7.370 \times 10^{1} \pm$ $3.32 \times 10^{0}$ | + | $\mathbf{-9.9 \times 10^{1} \pm}$ $\mathbf{0.00 \times 10^{0}}$ |
| fu34 | $5.331 \times 10^{10} \pm$ $3.51 \times 10^{10}$ | $8.041 \times 10^{9} \pm$ $4.88 \times 10^{9}$ | $2.476 \times 10^{2} \pm$ $2.13 \times 10^{3}$ | $4.896 \times 10^{5} \pm$ $2.21 \times 10^{5}$ | + | $\mathbf{1.790 \times 10^{0} \pm}$ $\mathbf{2.67 \times 10^{0}}$ |
| fu35 | $9.384 \times 10^{2} \pm$ $1.78 \times 10^{2}$ | $5.578 \times 10^{2} \pm$ $8.53 \times 10^{1}$ | $1.612 \times 10^{3} \pm$ $2.32 \times 10^{2}$ | $6.701 \times 10^{2} \pm$ $6.36 \times 10^{1}$ | + | $\mathbf{1.046 \times 10^{-2} \pm}$ $\mathbf{1.66 \times 10^{-2}}$ |
| fu36 | $7.462 \times 10^{2} \pm$ $2.32 \times 10^{2}$ | $2.422 \times 10^{2} \pm$ $8.72 \times 10^{1}$ | $\mathbf{-1.273 \times 10^{2} \pm}$ $\mathbf{3.77 \times 10^{0}}$ | $-1.082 \times 10^{2} \pm$ $4.21 \times 10^{0}$ | − | $3.137 \times 10^{-2} \pm$ $6.55 \times 10^{-2}$ |
| fu37 | $5.5078 \times 10^{2} \pm$ $1.83 \times 10^{-1}$ | $5.478 \times 10^{2} \pm$ $9.65 \times 10^{-1}$ | $5.498 \times 10^{2} \pm$ $4.64 \times 10^{-2}$ | $5.492 \times 10^{2} \pm$ $2.51 \times 10^{-1}$ | + | $\mathbf{6.525 \times 10^{-2} \pm}$ $\mathbf{4.01 \times 10^{-2}}$ |
| fu38 | $-1.201 \times 10^{3} \pm$ $4.77 \times 10^{1}$ | $\mathbf{-1.407 \times 10^{3} \pm}$ $\mathbf{3.24 \times 10^{1}}$ | $-1.267 \times 10^{3} \pm$ $5.18 \times 10^{1}$ | $-1.284 \times 10^{3} \pm$ $3.90 \times 10^{1}$ | − | $0.000 \times 10^{0} \pm$ $0.00 \times 10^{0}$ |
| fu39 | $6.157 \times 10^{4} \pm$ $1.54 \times 10^{4}$ | $4.98 \times 10^{-27} \pm$ $4.22 \times 10^{-27}$ | $\mathbf{1.445 \times 10^{-30} \pm}$ $\mathbf{5.58 \times 10^{-31}}$ | $4.314 \times 10^{-3} \pm$ $1.24 \times 10^{-3}$ | − | $8.704 \times 10^{-3} \pm$ $2.16 \times 10^{-4}$ |

**Table 3.** *Cont.*

| Function | rCGA | cDE | ISPO | cPSO | W | SSPCCSO |
|---|---|---|---|---|---|---|
| fu40 | $7.518 \times 10^4 \pm$ $1.08 \times 10^4$ | $3.316 \times 10^4 \pm$ $8.12 \times 10^3$ | $5.665 \times 10^2 \pm$ $2.19 \times 10^2$ | $4.375 \times 10^0 \pm$ $9.83 \times 10^{-1}$ | + | $\mathbf{1.463 \times 10^{-1}}$ $\mathbf{\pm 2.02 \times 10^{-1}}$ |
| fu41 | $1.044 \times 10^{10} \pm$ $4.34 \times 10^9$ | $1.098 \times 10^3 \pm$ $1.86 \times 10^3$ | $2.575 \times 10^2 \pm$ $3.11 \times 10^2$ | $8.941 \times 10^1 \pm$ $5.26 \times 10^1$ | − | $\mathbf{1.397 \times 10^0 \pm}$ $\mathbf{5.03 \times 10^0}$ |
| fu42 | $1.949 \times 10^1 \pm$ $2.59 \times 10^{-1}$ | $8.003 \times 10^0 \pm$ $4.31 \times 10^0$ | $1.949 \times 10^1 \pm$ $1.48 \times 10^{-1}$ | $1.277 \times 10^0 \pm$ $3.68 \times 10^{-1}$ | + | $\mathbf{7.812 \times 10^{-2}}$ $\mathbf{\pm 3.72 \times 10^{-2}}$ |
| fu43 | $2.978 \times 10^{-1} \pm$ $3.723 \times 10^{-1}$ | $1.354 \times 10^{-1} \pm$ $2.31 \times 10^{-1}$ | $6.857 \times 10^0 \pm$ $1.06 \times 10^1$ | $1.084 \times 10^0 \pm$ $3.16 \times 10^{-1}$ | + | $\mathbf{2.039 \times 10^{-2} \pm}$ $\mathbf{3.72 \times 10^{-2}}$ |
| fu44 | $4.707 \times 10^{-3} \pm$ $7.38 \times 10^{-3}$ | $\mathbf{0.001 \times 10^0 \pm}$ $\mathbf{0.01 \times 10^0}$ | $\mathbf{0.001 \times 10^0 \pm}$ $\mathbf{0.01 \times 10^0}$ | $\mathbf{0.001 \times 10^0 \pm}$ $\mathbf{0.01 \times 10^0}$ | = | $\mathbf{0.001 \times 10^0 \pm}$ $\mathbf{0.01 \times 10^0}$ |
| fu45 | $4.258 \times 10^4 \pm$ $4.15 \times 10^4$ | $2.534 \times 10^4 \pm$ $6.28 \times 10^3$ | $4.066 \times 10^3 \pm$ $9.66 \times 10^2$ | $5.051 \times 10^1 \pm$ $\mathbf{4.28 \times 10^1}$ | + | $\mathbf{1.433 \times 10^1 \pm}$ $\mathbf{1.92 \times 10^1}$ |
| fu46 | $2.368 \times 10^4 \pm$ $3.45 \times 10^3$ | $2.01 \times 10^4 \pm$ $3.04 \times 10^3$ | $3.776 \times 10^4 \pm$ $6.47 \times 10^3$ | $2.320 \times 10^4 \pm$ $3.38 \times 10^3$ | + | $\mathbf{3.577 \times 10^3 \pm}$ $\mathbf{8.68 \times 10^3}$ |
| fu47 | $2.087 \times 10^6 \pm$ $7.97 \times 10^5$ | $4.588 \times 10^5 \pm$ $1.69 \times 10^5$ | $\mathbf{1.589 \times 10^4 \pm}$ $\mathbf{1.74 \times 10^4}$ | $1.395 \times 10^6 \pm$ $1.14 \times 10^6$ | − | $3.749 \times 10^6 \pm$ $2.47 \times 10^6$ |

### 4.3. Comparison between the Corresponding Population-Based Algorithms and SSPCCSO

In addition to the comparison with memory saving algorithms, another group of comparisons between memory-saving and non-memory-saving algorithms are also used to test the performance of the algorithms; The comparison between sspccso, CSO, PSO and de will be arranged in this group of experiments. Table 4 presents the comparison results of the 47 test functions. SSPCCSO did better in 10 benchmarks, even when only one cat was employed. This status also happens in comparison between DE [23], cDE [2], PSO [24] and cPSO [4]. Obviously, Compared with population-based bio-inspired algorithms, the search ability of a single individual in SPCCSO is limited, and there is no large population collective cooperative search. However, the performance of SPCCSO still exceeds other algorithms.

**Table 4.** Comparison among SSPCCSO, CSO, PSO and DE.

| Benmark | DE | PSO | W | CSO | W | SSPCCSO |
|---|---|---|---|---|---|---|
| fu1 | $8.269 \times 10^1 \pm$ $1.90 \times 10^1$ | $1.095 \times 10^4 \pm$ $2.30 \times 10^3$ | − | $\mathbf{0.000 \times 10^0 \pm}$ $\mathbf{0.00 \times 10^0}$ | − | $6.170 \times 10^{-3} \pm$ $1.05 \times 10^{-3}$ |
| fu2 | $3.063 \times 10^4 \pm$ $3.70 \times 10^3$ | $4.232 \times 10^4 \pm$ $1.84 \times 10^3$ | + | $\mathbf{0.000 \times 10^0 \pm}$ $\mathbf{0.00 \times 10^0}$ | − | $3.625 \times 10^2 \pm$ $7.02 \times 10^2$ |
| fu3 | $\mathbf{2.715 \times 10^0 \pm}$ $\mathbf{1.11 \times 10^6}$ | $1.103 \times 10^9 \pm$ $5.07 \times 10^8$ | + | $2.890 \times 10^1 \pm$ $1.394 \times 10^{-2}$ | − | $5.776 \times 10^{-1} \pm$ $7.01 \times 10^0$ |
| fu4 | $4.072 \times 10^1 \pm$ $1.98 \times 10^{-1}$ | $1.639 \times 10^1 \pm$ $1.21 \times 10^0$ | + | $\mathbf{0.001 \times 10^0 \pm}$ $\mathbf{0.00 \times 10^0}$ | − | $5.574 \times 10^{-1} \pm$ $3.07 \times 10^{-2}$ |
| fu5 | $7.195 \times 10^1 \pm$ $9.73 \times 10^0$ | $\mathbf{0.001 \times 10^0 \pm}$ $\mathbf{0.001 \times 10^0}$ | − | $0.001 \times 10^0 \pm$ $0.01 \times 10^0$ | − | $1.613 \times 10^{-2} \pm$ $2.37 \times 10^{-3}$ |
| fu6 | $2.151 \times 10^2 \pm$ $9.08 \times 10^0$ | $2.887 \times 10^2 \pm$ $3.28 \times 10^1$ | + | $\mathbf{0.001 \times 10^0 \pm}$ $\mathbf{0.00 \times 10^0}$ | − | $2.399 \times 10^{-2} \pm$ $4.09 \times 10^{-3}$ |
| fu7 | $2.408 \times 10^5 \pm$ $4.96 \times 10^4$ | $1.321 \times 10^5 \pm$ $1.03 \times 10^4$ | + | $\mathbf{2.990 \times 10^2 \pm}$ $\mathbf{0.00 \times 10^0}$ | = | $2.991 \times 10^2 \pm$ $1.75 \times 10^{-1}$ |
| fu8 | $6.328 \times 10^3 \pm$ $2.36 \times 10^2$ | $6.677 \times 10^3 \pm$ $6.44 \times 10^2$ | − | $3.161 \times 10^3 \pm$ $9.76 \times 10^2$ | − | $\mathbf{1.248 \times 10^0 \pm}$ $\mathbf{1.90 \times 10^0}$ |
| fu9 | $1.633 \times 10^4 \pm$ $1.13 \times 10^3$ | $1.305 \times 10^4 \pm$ $3.17 \times 10^3$ | + | $1.344 \times 10^4 \pm$ $1.74 \times 10^3$ | + | $1.111 \times 10^5 \pm$ $5.29 \times 10^4$ |
| fu10 | $\mathbf{8.508 \times 10^5 \pm}$ $\mathbf{9.24 \times 10^4}$ | $9.716 \times 10^5 \pm$ $1.57 \times 10^5$ | − | $3.222 \times 10^6 \pm$ $1.68 \times 10^5$ | + | $9.390 \times 10^5 \pm$ $1.13 \times 10^4$ |

<div align="center">

**Table 4.** *Cont.*

</div>

| Benmark | DE | PSO | W | CSO | W | SSPCCSO |
|---------|-----|-----|---|-----|---|---------|
| fu11 | $4.217 \times 10^0 \pm$ $1.58 \times 10^{-1}$ | $1.707 \times 10^1 \pm 1.73$ $\times 10^0$ | + | $\mathbf{-1.84 \times 10^{-6}} \pm$ $\mathbf{0.01 \times 10^0}$ | − | $8.328 \times 10^{-2} \pm$ $8.23 \times 10^{-2}$ |
| fu12 | $6.536 \times 10^1 \pm$ $1.02 \times 10^1$ | $1.139 \times 10^1 \pm$ $3.08 \times 10^1$ | + | $9.568 \times 10^{-8} \pm$ $2.68 \times 10^{-8}$ | = | $\mathbf{1.018 \times 10^{-8}} \pm$ $\mathbf{2.61 \times 10^9}$ |
| fu13 | $\mathbf{2.586 \times}$ $\mathbf{10^2 \pm 1.12 \times 10^1}$ | $3.155 \times 10^2 \pm$ $2.19 \times 10^1$ | + | $2.701 \times 10^2 \pm$ $0.01 \times 10^0$ | = | $2.70 \times 10^2 \pm$ $1.81 \times 10^{-5}$ |
| fu14 | $4.003 \times 10^1 \pm$ $1.09 \times 10^0$ | $\mathbf{3.966 \times 10^1 \pm}$ $\mathbf{1.18 \times 10^0}$ | − | $7.049 \times 10^2 \pm$ $2.22 \times 10^0$ | = | $7.142 \times 10^2 \pm$ $2.37 \times 10^{-1}$ |
| fu15 | $7.443 \times 10^{-2} \pm$ $1.89 \times 10^{-5}$ | $4.083 \times 10^0 \pm$ $2.23 \times 10^0$ | + | $0.001 \times 10^0 \pm$ $0.01 \times 10^0$ | − | $9.427 \times 10^{-3} \pm$ $6.15 \times 10^{-3}$ |
| fu16 | $-9.942 \times 10^{-8} \pm$ $1.08 \times 10^{-1}$ | $-1.001 \times 10^2 \pm$ $0.01 \times 10^0$ | = | $-1.001 \times 10^2 \pm$ $8.46 \times 10^{-5}$ | = | $\mathbf{-1.000 \times 10^2 \pm}$ $\mathbf{0.00 \times 10^0}$ |
| fu17 | $\mathbf{9.424 \times 10^{-8} \pm}$ $\mathbf{5.16 \times 10^{-8}}$ | $1.046 \times 10^1 \pm$ $5.08 \times 10^0$ | − | $1.631 \times 10^0 \pm$ $5.84 \times 10^{-1}$ | − | $9.518 \times 10^{-5} \pm$ $1.57 \times 10^{-6}$ |
| fu18 | $\mathbf{-1.151 \times 10^0 \pm}$ $\mathbf{3.37 \times 10^{-7}}$ | $3.502 \times 10^3 \pm$ $9.85 \times 10^3$ | + | $5454 \times 10^{-1} \pm$ $3.27 \times 10^{-1}$ | + | $-4.104 \times 10^{-1} \pm$ $8.97 \times 10^{-4}$ |
| fu19 | $4.701 \times 10^2 \pm$ $1.44 \times 10^1$ | $6.107 \times 10^2 \pm$ $3.43 \times 10^1$ | + | $\mathbf{4.501 \times 10^2 \pm}$ $\mathbf{0.01 \times 10^0}$ | = | $\mathbf{4.500 \times 10^2 \pm}$ $\mathbf{2.60 \times 10^{-3}}$ |
| fu20 | $-1.278 \times 10^1 \pm$ $4.28 \times 10^{-1}$ | $-1.936 \times 10^1 \pm$ $1.72 \times 10^0$ | + | $-1.103 \times 10^1 \pm$ $1.08 \times 10^0$ | + | $\mathbf{-1.988 \times 10^1 \pm}$ $\mathbf{2.33 \times 10^{-1}}$ |
| fu21 | $\mathbf{1.268 \times 10^4 \pm}$ $\mathbf{3.62 \times 10^2}$ | $9.691 \times 10^3 \pm$ $1.14 \times 10^3$ | − | $4.785 \times 10^3 \pm$ $1.48 \times 10^2$ | − | $1.42 \times 10^0 \pm$ $2.26 \times 10^0$ |
| fu22 | $1.828 \times 10^1 \pm$ $4.24 \times 10^1$ | $2.004 \times 10^1 \pm$ $3.75 \times 10^{-1}$ | + | $\mathbf{0.001 \times 10^0 \pm}$ $\mathbf{0.01 \times 10^0}$ | − | $7.139 \times 10^{-2} \pm$ $4.17 \times 10^{-2}$ |
| fu23 | $1.611 \times 10^2 \pm$ $6.39 \times 10^0$ | $1.815 \times 10^1 \pm$ $1.01 \times 10^1$ | + | $\mathbf{4.658 \times 10^{-2} \pm}$ $\mathbf{2.38 \times 10^{-2}}$ | − | $7.319 \times 10^{-2} \pm$ $5.17 \times 10^{-3}$ |
| fu24 | $2.386 \times 10^4 \pm$ $3.48 \times 10^3$ | $6.501 \times 10^4 \pm$ $9.68 \times 10^3$ | + | $\mathbf{0.001 \times 10^0 \pm}$ $\mathbf{0.01 \times 10^0}$ | − | $8.961 \times 10^{-3} \pm$ $1.46 \times 10^{-2}$ |
| fu25 | $\mathbf{-1.119 \times 10^{-2} \pm}$ $\mathbf{1.29 \times 10^{-3}}$ | $-7.493 \times 10^{-3} \pm$ $1.06 \times 10^{-3}$ | − | $0.001 \times 10^0 \pm$ $0.01 \times 10^0$ | = | $0.000 \times 10^0 \pm$ $0.00 \times 10^0$ |
| fu26 | $-1.589 \times 10^1 \pm$ $5.26 \times 10^{-1}$ | $-2.677 \times 10^1 \pm$ $2.01 \times 10^0$ | − | $-1.978 \times 10^1 \pm$ $1.43 \times 10^0$ | = | $-3.970 \times 10^1 \pm$ $1.52 \times 10^{-1}$ |
| fu27 | $8.899 \times 10^4 \pm$ $8.79 \times 10^3$ | $1.925 \times 10^4 \pm$ $1.93 \times 10^4$ | + | $\mathbf{0.001 \times 10^0 \pm}$ $\mathbf{0.01 \times 10^0}$ | − | $1.033 \times 10^{-2} \pm$ $1.39 \times 10^{-2}$ |
| fu28 | $1.177 \times 10^3 \pm$ $2.54 \times 10^1$ | $1.279 \times 10^3 \pm 4.45$ $\times 10^1$ | + | $\mathbf{0.001 \times 10^0 \pm}$ $\mathbf{0.01 \times 10^0}$ | − | $6.631 \times 10^{-3} \pm$ $1.24 \times 10^{-4}$ |
| fu29 | $2.636 \times 10^{10} \pm$ $5.09 \times 10^9$ | $3.854 \times 10^{10} \pm$ $1.42 \times 10^{10}$ | + | $\mathbf{9.899 \times 10^1 \pm}$ $\mathbf{1.85 \times 10^0}$ | − | $1.286 \times 10^0 \pm$ $2.50 \times 10^0$ |
| fu30 | $1.477 \times 10^{11} \pm$ $1.27 \times 10^{10}$ | $1.017 \times 10^{11} \pm$ $1.96 \times 10^{10}$ | + | $\mathbf{0.001 \times 10^0 \pm}$ $\mathbf{8.36 \times 10^0}$ | − | $1.109 \times 10^8 \pm$ $3.44 \times 10^8$ |
| fu31 | $3.026 \times 10^4 \pm$ $4.79 \times 10^2$ | $2.368 \times 10^4 \pm$ $1.89 \times 10^3$ | − | $\mathbf{1.046 \times 10^4 \pm}$ $\mathbf{2.95 \times 10^3}$ | − | $4.989 \times 10^4 \pm$ $8.38 \times 10^4$ |
| fu32 | $2.094 \times 10^5 \pm$ $1.64 \times 10^4$ | $1.138 \times 10^4 \pm$ $1.68 \times 10^3$ | + | $\mathbf{1.532 \times 10^{-2} \pm}$ $\mathbf{3.81 \times 10^{-3}}$ | − | $1.774 \times 10^{-2} \pm$ $2.89 \times 10^{-2}$ |
| fu33 | $1.186 \times 10^2 \pm$ $2.84 \times 10^0$ | $1.407 \times 10^2 \pm$ $1.28 \times 10^1$ | + | $7.371 \times 10^3 \pm$ $3.33 \times 10^0$ | + | $\mathbf{-9.9 \times 10^1 \pm}$ $\mathbf{0.00 \times 10^0}$ |
| fu34 | $8.197 \times 10^{10} \pm$ $1.12 \times 10^{10}$ | $7.765 \times 10^{10} \pm$ $2.17 \times 10^{10}$ | + | $9.898 \times 10^2 \pm$ $2.25 \times 10^{-2}$ | + | $\mathbf{1.790 \times 10^0 \pm}$ $\mathbf{2.67 \times 10^0}$ |
| fu35 | $1.398 \times 10^3 \pm$ $4.25 \times 10^1$ | $1.055 \times 10^3 \pm$ $1.48 \times 10^2$ | + | $\mathbf{0.001 \times 10^0 \pm}$ $\mathbf{0.01 \times 10^0}$ | − | $1.046 \times 10^{-2} \pm$ $1.66 \times 10^{-2}$ |
| fu36 | $1.567 \times 10^3 \pm$ $1.34 \times 10^2$ | $1.242 \times 10^3 \pm$ $2.45 \times 10^2$ | + | $1.083 \times 10^2 \pm$ $4.22 \times 10^0$ | + | $\mathbf{3.137 \times 10^{-2} \pm}$ $\mathbf{6.55 \times 10^{-2}}$ |
| fu37 | $5.507 \times 10^2 \pm$ $1.25 \times 10^{-1}$ | $5.508 \times 10^2 \pm$ $1.71 \times 10^{-1}$ | + | $5.493 \times 10^2 \pm 2.51$ $\times 10^{-1}$ | + | $\mathbf{6.525 \times 10^{-2} \pm}$ $\mathbf{4.01 \times 10^{-2}}$ |

**Table 4.** *Cont.*

| Benmark | DE | PSO | W | CSO | W | SSPCCSO |
|---|---|---|---|---|---|---|
| fu38 | $-1.056 \times 10^3 \pm$ $1.09 \times 10^1$ | $-1.283 \times 10^3 \pm$ $2.18 \times 10^2$ | − | $\mathbf{-1.285 \times}$ $\mathbf{10^3 \pm 3.91 \times}$ $\mathbf{10^1}$ | − | $0.000 \times 10^0 \pm$ $0.00 \times 10^0$ |
| fu39 | $8.338 \times 10^3 \pm$ $1.12 \times 10^3$ | $1.201 \times 10^3 \pm$ $2.18 \times 10^2$ | + | $\mathbf{0.001 \times 10^0 \pm}$ $\mathbf{0.01 \times 10^0}$ | − | $8.704 \times 10^{-3} \pm$ $2.16 \times 10^{-4}$ |
| fu40 | $8.969 \times 10^4 \pm$ $7.75 \times 10^3$ | $1.701 \times 10^4 \pm$ $3.08 \times 10^3$ | + | $\mathbf{0.001 \times 10^0 \pm}$ $\mathbf{0.01 \times 10^0}$ | − | $1.463 \times 10^{-1} \pm$ $2.02 \times 10^{-1}$ |
| fu41 | $2.142 \times 10^9 \pm$ $5.71 \times 10^8$ | $1.784 \times 10^7 \pm$ $5.52 \times 10^6$ | + | $4.898 \times 10^1 \pm$ $1.38 \times 10^{-2}$ | − | $\mathbf{1.397 \times 10^0 \pm}$ $\mathbf{5.03 \times 10^0}$ |
| fu42 | $1.364 \times 10^1 \pm$ $4.46 \times 10^{-1}$ | $6.878 \times 10^0 \pm$ $4.73 \times 10^{-1}$ | + | $\mathbf{0.001 \times 10^0 \pm}$ $\mathbf{0.01 \times 10^0}$ | − | $7.812 \times 10^{-2} \pm$ $3.72 \times 10^{-2}$ |
| fu43 | $3.711 \times 10^{-2} \pm$ $3.27 \times 10^1$ | $2.555 \times 10^{-2} \pm$ $4.98 \times 10^1$ | − | $\mathbf{0.001 \times 10^0 \pm}$ $\mathbf{0.01 \times 10^0}$ | − | $2.039 \times 10^{-2} \pm$ $3.72 \times 10^{-2}$ |
| fu44 | $4.684 \times 10^2 \pm$ $1.35 \times 10^1$ | $\mathbf{0.001 \times 10^0 \pm}$ $\mathbf{0.01 \times 10^0}$ | = | $\mathbf{0.001 \times 10^0 \pm}$ $\mathbf{0.01 \times 10^0}$ | = | $\mathbf{0.001 \times 10^0 \pm}$ $\mathbf{0.01 \times 10^0}$ |
| fu45 | $2.539 \times 10^6 \pm$ $2.09 \times 10^5$ | $1.134 \times 10^6 \pm$ $2.17 \times 10^3$ | + | $\mathbf{0.001 \times 10^0 \pm}$ $\mathbf{0.01 \times 10^0}$ | − | $1.433 \times 10^1 \pm$ $1.92 \times 10^1$ |
| fu46 | $3.152 \times 10^4 \pm$ $1.18 \times 10^3$ | $1.888 \times 10^4 \pm$ $2.17 \times 10^3$ | + | $\mathbf{0.0010 \times 10^0 \pm}$ $\mathbf{0.01 \times 10^0}$ | − | $3.577 \times 10^3 \pm$ $8.68 \times 10^3$ |
| fu47 | $4.675 \times 10^6 \pm$ $2.19 \times 10^5$ | $\mathbf{2.183 \times 10^6 \pm}$ $\mathbf{4.01 \times 10^5}$ | − | $1.694 \times 10^7 \pm$ $4.31 \times 10^6$ | + | $3.749 \times 10^6 \pm$ $2.47 \times 10^6$ |

### 4.4. Comparison against Swarm-Based Version Algorithms Based on Iterations and Solution

Another indicator of the algorithm's performance is the convergence rate; after several iterations, different algorithms will converge to different results. A comparison of the convergence results based on the same test function and the same number of iterations is shown in Table 5. Test function 1 [22] is selected.

**Table 5.** The convergence results on test function 1 based on the same iterations.

| Iterations | PSO | CSO | cPSO | SSPCCSO |
|---|---|---|---|---|
| I 100 | 8202.6317 | 0.000000 | 55.695 | 5.809 |
| I 200 | 3754.9483 | 0.000000 | 19.852 | 4.277 |
| I 1000 | 1417.4688 | 0.000000 | 0.61648 | 0.03692 |
| I 2000 | 1414.2868 | 0.000000 | 0.36894 | 0.000032 |

Table 5 shows that the SSPCCSO exceeds the other compared algorithms with a faster convergence rate, except for CSO. It ensures gradual convergence in the early iterations. Table 6 shows that the SSPCCSO is also much better than the PSO and CPSO in terms of the convergence results

**Table 6.** The convergence results on test function 4 based on the same iterations.

| Iterations | PSO | CSO | cPSO | SSPCCSO |
|---|---|---|---|---|
| I 100 | 14.38578 | $8.88 \times 10^{-16}$ | 6.3458 | 10.602 |
| I 200 | 12.03479 | $8.88 \times 10^{-16}$ | 4.6725 | 4.942 |
| I 500 | 10.31688 | $8.88 \times 10^{-16}$ | 3.3370 | 3.824 |
| I 1000 | 9.05237 | $8.88 \times 10^{-16}$ | 2.3393 | 1.427 |

Due to space constraints, no more comparison results are displayed. This situation also could be obtained with other test functions results.

Because of too many local cycles in the seeking mode, there is not any advantage shown in the computing costs. However, the gradient descent method introduced can make up for this shortcoming. It can end unnecessary calculations in advance, so the algorithm achieves a better performance in less running time.

## 5. Conclusions

A novel compact cat swarm optimization scheme based on gradient descent is proposed in this study. It kept the search logic of CSO, but introduced a gradient descent method into the scheme to seek for the optimal solution. According to the experimental results, this scheme could greatly reduce the computing costs. It also outperformed all of the relative compact optimization algorithms in most of the test benchmark functions. More significantly, its design is based on gamma probability distribution for solving small size sampling problems, so it probably suggests a new solution for optimization problems with small sampling.

**Author Contributions:** Conceptualization, M.Z.; software, Z.H. Resources, T.L.; data curation, Y.Y.; writing—original draft preparation, M.Z.; writing—review and editing, Z.H.; funding acquisition, M.Z. All authors have read and agreed to the published version of the manuscript.

**Funding:** This research was supported by Hubei Provincial Department of Education: 21D031.

**Institutional Review Board Statement:** Not applicable.

**Informed Consent Statement:** Not applicable.

**Data Availability Statement:** Not applicable.

**Conflicts of Interest:** The authors declare no conflict of interest.

## Appendix A

(1) Test function 1:

$$f_1(\mathrm{y}) = \sum_{i=1}^{D} z_i^2 z_i = \mathrm{y} - o; \quad D = [-100, 100]^{30}$$

(2) Test function 2:

$$f_2(\mathrm{y}) = \sum_{i=1}^{D} (\sum_{j=1}^{i} \mathrm{x}_j)^2 z_i = x - o; \quad D = [-100, 100]^{30}$$

(3) Test function 3:

$$f_3(x) = \sum_{i=1}^{n-1} [100(x_{i+1} - x_i^2)^2 + (x_i - 1)^2], D = [-100, 100]^{30}$$

(4) Test function 4:

$$f_4(x) = -20e^{-0.2\sqrt{\frac{1}{n}\sum_{i=1}^{n} z_i}} - e^{\frac{1}{n}\sum_{i=1}^{n} \cos(2*pi*z_i)} + 20 + e, z_i = x - o; \quad D = [-32, 32]^{30}$$

(5) Test function 5:

$$f_5(x) = \sum_{i=1}^{n} \frac{z_i^2}{4000} - \prod_{i=1}^{n} \cos(\frac{z_i}{\sqrt{i}}) + 1, z_i = x - o; D = [-600, 600]^{30},$$

(6) Test function 6:

$$f_6(x) = 10n + \sum_{i=1}^{n} [z_i^2 - 10\cos(2\pi z_i)], \ z_i = x - o, o = [o_1, o_2, o_3, \ldots o_n], D = [-5,5]^{30}$$

(7) Test function 7:

$$f_7(x) = \sum_{i=1}^{M} [y_i^2 - 10\cos(2\pi y_i)] + 10n$$

$$y_i = \begin{cases} z_i & if|z_i| < 1/2 \\ round(2z_i)/2 & if|z_i| > 1/2 \end{cases} \quad z_i = x - o; \ D = [-500,500]^{30}$$

(8) Test function 8:

$$f_8(x) = 418.9829n + \sum_{i=1}^{n} (-x_i \sin|x_i|), \ D = [-500,500]^{30}$$

(9) Test function 9:

$$f_9(x) = max_i(|A_i x_i - B_i|), \ B_i = A_i \times o_i, D = [-100,100]^{30}$$

(10) Test function 10:

$$f_{10}(x) = \sum_{i=1}^{n} (A_i x_i - B_i(x))^2$$

(11) Test function 11:

$$f_{11}(x) = -20e^{-0.2\sqrt{\frac{1}{n}\sum_{i=1}^{n} z_i}} - e^{\frac{1}{n}\sum_{i=1}^{n}\cos(2*pi*z_i)} + 20 + e, \ z_i = M(x-o), Cond(M) = 1 \quad, D = [-32,32]^{30}$$

(12) Test function 12:

$$f_{12}(x) = \sum_{i=1}^{n} \frac{z_i^2}{4000} - \prod_{i=1}^{n} \cos(\frac{z_i}{\sqrt{i}}) + 1$$

$$z_i = M(x-o), Cond(M) = 3, o = [o_1, o_2, o_3, \ldots o_n], D = [-600,600]^{30}$$

(13) Test function 13:

$$f_{13}(x) = 10n + \sum_{i=1}^{M} [z_i^2 - 10\cos(2\pi z_i)]$$

$$z_i = M(x-o), Cond(M) = 3, o = [o_1, o_2, o_3, \ldots o_n], D = [-5,5]^{30}$$

(14) Test function 14:

$$o = [o_1, o_2, o_3, \ldots o_n]$$

$$f_{14}(x) = \sum_{i=1}^{n}\sum_{k=0}^{k_{max}} (a^k \cos(2\pi b^k(z_i + 0.5))) - n\sum_{k=0}^{k_{max}} a^k \cos(2\pi b^k) * 0.5$$

$$a = 0.5, b = 0.3, k_{max} = 20, z = M(x-o), M = 5, D = [-0.5, 0.5]^{30}$$

(15) Test function 15:

$$f_{15}(x) = \sum_{i=1}^{n} |x_i| - \prod_{i=1}^{n} |x_i|, \ D = [-10,10]^{10}$$

(16) Test function 16:

$$f_{16}(x) = \max_{i=1}^{n}|x_i|, \ D = [-100,100]^{10}$$

(17) Test function 17:

$$f_{17}(x) = \frac{\pi}{n}\left\{10 * \sin^2 \pi y_1 + \sum_{i=1}^{n} \left((y_i - 1)^2(1 + 10\sin^2 \pi y_i)\right) + (y_n - 1)^2\right\} + \sum_{i=1}^{n} u(x_i, 10, 100, 4)$$

$$\text{where } y_i = 1 + \frac{1}{4}(x_i + 1), \text{ and } u(x, a, k, m) = \begin{cases} k(x_i - a)^m & if\, x_i > a \\ 0 & f\,|x_i| \leq a \\ k(-x_i - a)^m & if\, x_i < -a \end{cases}$$

$$D = [-50, 50]^{10}$$

(18) Test function 18:

$$f_{18}(x) = \frac{1}{10}\left\{\sin^2 3\pi x_1 + \sum_{i=1}^{n-1}\left((x_i - 1)^2(1 + \sin^2 3\pi x_{i+1})\right)\right\} + \frac{1}{10}\left\{(x_n - 1)(1 + \sin 2\pi x_n)^2\right\} + \sum_{i=1}^{n} u(x_i, 5, 100, 4)$$

$$\text{where } D = [-50, 50]^{10}$$

(19) Test function 19:

$$f_{19}(x) = 10n + \sum_{i=1}^{n} [z_i^2 - 10\cos(2\pi z_i)],\ z_i = x - o;\ D = [-5, 5]^{50}$$

(20) Test function 20:

$$f_{20}(x) = -\sum_{i=1}^{n} \sin(x_i)[\sin(\frac{ix_i^2}{\pi})]^{2m},\ m = 10, D = [0, \pi]^{50}$$

(21) Test function 21:

$$f_{21}(x) = 418.9829n + \sum_{i=1}^{n} \left(-x_i \sin(\sqrt{|x_i|})\right), D = [-500, 500]^{30}$$

(22) Test function 22:

$$f_{22}(x) = -20e^{-0.2\sqrt{\frac{1}{n}\sum_{i=1}^{n} z_i}} - e^{\frac{1}{n}\sum_{i=1}^{n}\cos(2*pi*z_i)} + 20 + e, z_i = x - o;\ D = [-32, 32]^{100}$$

(23) Test function 23:

$$f_{23}(x) = \prod_{i=1}^{n} \sin(x_i)\sqrt{\prod_{i=1}^{n}(x_i)},\ D = [-10, 10]^{100}$$

(24) Test function 24:

$$f_{24}(x) = \sum_{i=1}^{n} (i \cdot x_i^2),\ D = [-10, 10]^{100}$$

(25) Test function 25:

$$f_{25}(x) = -\frac{1 + \cos(12\sqrt{\sum_{i=1}^{n} x_i^2})}{\frac{1}{2}\sum_{i=1}^{n} x_i^2 + 2}, D = [-5.12, 5.12]^{100}$$

(26) Test function 26:

$$f_{26}(x) = -\sum_{i=1}^{n} \sin(x_i)[\sin(\frac{ix_i^2}{\pi})]^{2m}, \ m = 10, D = [0, \pi]^{100}$$

(27) Test function 27:

$$f_{27}(x) = \sum_{i=1}^{n} 5i \cdot x_i^2, \ D = [-5.12, 5.12]^{100}$$

(28) Test function 28:

$$f_{28}(x) = 10n + \sum_{i=1}^{n} [z_i^2 - 10\cos(2\pi z_i)], \ z_i = x - o; \ D = [-5.12, 5.12]^{100}$$

(29) Test function 29:

$$f_{29}(x) = \sum_{i=1}^{n-1} [100(x_{i+1} - x_i^2)^2 + (x_i - 1)^2], \ D = [-100, 100]^{100}$$

(30) Test function 30:

$$f_{30}(x) = \sum_{i=1}^{n} \sum_{j=1}^{i} x_j^2, D = [-65536, \ 65536]^{100}$$

(31) Test function 31:

$$f_{31}(x) = 418.9829n + \sum_{i=1}^{n} -x_i \sin(\sqrt{|x_i|}), \ D = [-500, \ 500]^{100}$$

(32) Test function 32:

$$f_{32}(x) = \sum_{i=1}^{D} z_i^2, \ z_i = x - o; D = [-5, 5]^{100}$$

(33) Test function 33:

$$f_{33}(x) = \max_i |z_i|, \ z_i = x - o; \quad D = [-100, 100]^{100}$$

(34) Test function 34:

$$f_{34}(x) = \sum_{i=1}^{n-1} [100(x_{i+1} - x_i^2)^2 + (x_i - 1)^2], \ D = [-100, 100]^{100}$$

(35) Test function 35:

$$f_{35}(x) = 10n + \sum_{i=1}^{n} [z_i^2 - 10\cos(2\pi z_i)], \ z_i = x - o, o = [o_1, o_2, o_3, \ldots o_n], D = [-5, 5]^{30}$$

(36) Test function 36:

$$f_{36}(x) = \frac{1}{4000} \sum_{i=1}^{n} z_i^2 - \prod_{i=1}^{n} \cos(\frac{z_i}{\sqrt{i}}) + 1, \ z_i = x - o, o = [o_1, o_2, o_3, \ldots o_n], D = [-600, 600]^{100}$$

(37) Test function 37:

$$f_{37}(x) = -20e^{-0.2\sqrt{\frac{1}{n}\sum_{i=1}^{n} z_i}} - e^{\frac{1}{n}\sum_{i=1}^{n}\cos(2*pi*z_i)} + 20 + e,\ z_i = x - o, o = [o_1, o_2, o_3, \ldots o_n], D = [-5, 5]^{100}$$

(38) Test function 38:

$$f_{38}(x) = \sum_{i=1}^{n} fractal1D(x_i + twist(x_{(i\bmod n)+1}))$$
$$twist(x) = 4(x^4 - 2x^3 + x^2)$$
$$fractal1D(x) \approx \sum_{k=1}^{3} \sum_{1}^{2^{k-1}} \sum_{1}^{ran2(0)} doubledip(x, ran1(0), \tfrac{1}{2^{k-1}(2-ran1(0))})$$
$$doubledip(x, c, s) = \begin{cases} (-6144(x-c)^6 - 3088(x-c)^4 - 392(x-c)^2 + 1)s & 0.5 < x < 0.5 \\ 0 & otherwise \end{cases}$$
$$D = [-1, 1]^{100}$$

(39) Test function 39:

$$f_{39}(x) = \sum_{i=1}^{D} z_i^2,\ z_i = x - o, o = [o_1, o_2, o_3, \ldots o_n], D = [-100, 100]^{50}$$

(40) Test function 40:

$$f_{40}(x) = \sum_{i=1}^{n} (\sum_{j}^{i} z_i)^2,\ z_i = x - o, o = [o_1, o_2, o_3, \ldots o_n], D = [-100, 100]^{50}$$

(41) Test function 41:

$$f_{41}(x) = \sum_{i=1}^{n-1} [100(x_{i+1} - x_i^2)^2 + (x_i - 1)^2], D = [-100, 100]^{50}$$

(42) Test function 42:

$$f_{42}(x) = -20e^{-0.2\sqrt{\frac{1}{n}\sum_{i=1}^{n} z_i}} - e^{\frac{1}{n}\sum_{i=1}^{n}\cos(2*pi*z_i)} + 20 + e,\ z_i = x - o, o = [o_1, o_2, o_3, \ldots o_n], D = [-32, 32]^{50}$$

(43) Test function 43:

$$f_{43}(x) = \sum_{i=1}^{n} \frac{z_i^2}{4000} - \prod_{i=1}^{n} \cos(\frac{z_i}{\sqrt{i}}) + 1,\ z_i = x - o, o = [o_1, o_2, o_3, \ldots o_n], D = [-600, 600]^{50}$$

(44) Test function 44:

$$f_{44}(x) = 10n + \sum_{i=1}^{n} [z_i^2 - 10\cos(2\pi z_i)]$$
$$z_i = x - o, o = [o_1, o_2, o_3, \ldots o_n], D = [-5, 5]^{50}$$

(45) Test function 45:

$$f_{45}(x) = \sum_{i=1}^{M} [y_i^2 - 10\cos(2\pi y_i)] + 10n$$
$$y_i = \begin{cases} z_i & if |z_i| < 1/2 \\ round(2z_i)/2 & if |z_i| > 1/2 \end{cases}$$
$$z_i = x - o, o = [o_1, o_2, o_3, \ldots o_n], D = [-500, 500]^{50}$$

(46) Test function 46:

$$f_{46}(x) = \sum_{i=1}^{n} (\sum_{j=1}^{i} x_j)^2, \; D = [-100, 100]^{50}$$

(47) Test function 47:

$$f_{47}(x) = \sum_{i=1}^{n} (A_i x_i - B_i(x))^2, \; D = [-\pi, \pi]^{50}$$

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
