# Peer review of "A Compact Cat Swarm Optimization Algorithm Based on Small Sample Probability Model"

_applsci, doi:10.3390/app12168209_

Round 1
Reviewer 1 Report
First of all, the manuscript requires an exhaustive English review. THe document is full of unaccurate expressions, and really badly written sentences that truly limits the understanding of the paper.
On the other hand, I do not consider that the approach proposed by the author is relevant enough to be considered in this journal, although I think much of this is due to the poor quality of the writing.
Author Response
Dear Editors and Reviewers:
Thank you for your letter and for the reviewers’ comments concerning our manuscript entitled “A Compact Cat Swarm Optimization Algorithm based on Small Sample Probability Model”. Those comments are all valuable and very helpful for revising and improving our paper, as well as the important guiding significance to our research. We have studied comments carefully and have made correction which we hope to meet with approval. Revised portions are marked in blue and red in the paper. The main corrections in the paper and the responds to the reviewer’s comments are as attached files。

Reviewer 2 Report
The paper discusses a modification of Cat Swarm Optimization. The authors modify the existing algorithm by introducing new elements like gamma probability distribution for sampling, gradient descent in the tracking mode of a cat, and differential operation in the seeking mode. The paper is written chaotically, and no proper introduction to the problem was given. Probably only the specialists who know different variants of CSO well can understand the article. The results seem interesting, but the manuscript requires many changes to make it readable for a reader with general knowledge of optimization procedures.
- Authors should introduce the CSO algorithm. The reader is surprised by some detailed step descriptions without general information about the algorithm, the steps, what was the reasoning for creating them, etc. The general description would take around half a page but help readers, especially people who do not know CSO.
- Similarly, we can find a sentence (line 49), "These Compact evolutionary algorithms employed a probability distribution to explicit represent the population of solutions.". But no explanation of how the distribution is used, and what, in general, is the compact evolutionary algorithm.
- Authors should read equations carefully and check them. Mathematics requires a rigorous approach.
o In eq (1), we find \mu and \delta, while in the following sentence, we read \sigma is a standard deviation. Is that mistake, or \sigma should be introduced in the equation and \delta explained?
o In eq (3), x_i is given as a lower index of d, while it is probably the variable of the integration.
- The authors wrote (line 136), "Then each position in SMP will be recalculated by a mutagenic operator, a dimension of designed variable x_i, could be selected to mutate, and the range of variation is decided by a random number which is up to the 20% of x_i. The mutation operation will be presented as formula (6)".
Firstly, why \delta x_i is bounded to 20% of the value of x_i? This implies that the elements on the edge of the domain (x_i near 1) can be moved much farther by mutation than those with x_i near zero. This introduces the preference for positions near the origin.
Secondly, can \delta x_i be a negative number?
- The original CSO mutation operator is based on the cat's position x_i and random \delta (eq. 6). Now in the paper, we have a winner and loser. Who are they? Are these the best and the worst position in the iteration, in all positions visited during optimization? Two randomly chosen elements of solution sets? If the looser and winner are far from position x_i of a cat, why do authors call it "search for a local best."
Earlier in line 116, we have a sentence, "When two individual are compared, the winner indicates the one with better fitness, and the other is loser." Are these the same elements looser and winner?
Figure 4 suggests the winner is always nearer the mean than the loser. I believe it is not always accurate as deciding which point is a winner or loser depends on the fit function, not the distribution used in optimization.
- How do authors derive a gradient for eq. 17? How many local solutions are used to assess the gradient?
- The results are given in a not acceptable way. A reader cannot compare given results, and they look like a bunch of random numbers. What are the values in Table 3? Are these the best x positions? or values of fit function? What was the task of the optimization? To find the position of a minimal point of a benchmark function?
When we look at one of the functions f5 we see in this row results:
f5; 6.434e-03±1.30e-02; 4.288e-03±1.37e-02; 1.123e+01 ± 1.75e+01; 9.636e-08±3.07e-08; - 1.613e-02±2.37e-03
How can a reader know that 9.636e-08±3.07e-08 is the best value in this row? How far SSPCCSO What is the function "compete" used in pseudo-code (Figure 7.)
- Authors should change Figure 2 a little. It is too similar to the figure in Wikipedia for gamma distribution https://en.wikipedia.org/wiki/Gamma_distribution#/media/File:Gamma_distribution_pdf.svg or needs to add an appropriate citation. On the other hand, it is easy to create a picture to show the properties of the gamma distribution
- Authors often use "standard variation". They probably mean standard deviation, and they should be aware that variation is not a synonym for deviation in the statistical domain.
- The document is full of language errors, to list a few:
o Line 200 -- Firstly, each dimension \mu_i and \sigma_i associated with the each dimension x_i of designed variable x will be initialized (\mu_i=1 and \sigma_i=\lambda).
o In many sentences, a capital letter is missing at the beginning, like in Line 379-- A novel compact cat swarm optimization scheme based on gradient descent was proposed in this study. it keeps the search logic of CSO …
o Line 375 -- Because of too many local loops in seeking mode, there is no obvious advantage on computation cost. But the introduced gradient descent method can offset the disadvantage.
o Scientific articles are usually written in the present tense, not past ones. For example, in the abstract, in the first line, we have: "In This paper, a compact cat swarm optimization algorithm based on Small Sample Probability Model(SSPCCSO)was proposed"
Author Response

(The authors gave the same response as above.)

Reviewer 3 Report
A well organised introduction presenting the documentation used and the already existing research.
Watch out the spelling, English phrase and the fonts, because not all text is written in the same font.
Maybe the characteristics in figure 1 could be a little more explained.
An interesting subject and an interesting idea for the research!
The writing in figures 3,4,5 could be increased, so that it is readable! One can ungroup the figure and work exactly on the text. Watch out the letters and the capitals, some capital letters appear in the middle of a phrase.
Author Response

(The authors gave the same response as above.)

Round 2
Reviewer 2 Report
The changes made improved the manuscript.